# The Prevalence, Genotype Distribution and Risk Factors of Human Papillomavirus in Tunisia: A National-Based Study

**DOI:** 10.3390/v14102175

**Published:** 2022-09-30

**Authors:** Monia Ardhaoui, Hejer Letaief, Emna Ennaifer, Souha Bougatef, Thelja Lassili, Rahima Bel Haj Rhouma, Emna Fehri, Kaouther Ouerhani, Ikram Guizani, Myriam Mchela, Karim Chahed, Mohamed Kouni Chahed, Mohamed Samir Boubaker, Nissaf Bouafif Ben Alaya

**Affiliations:** 1Department of Human and Experimental Pathology, Institut Pasteur de Tunis, Tunis 1002, Tunisia; 2Laboratory of Molecular Epidemiology and Experimental Pathology, Institut Pasteur de Tunis, Tunis 1002, Tunisia; 3National Observatory of New and Emergent Diseases, Tunis 1002, Tunisia; 4Laboratory of Biomedical Genomics and Oncogenetics, Institut Pasteur de Tunis, Tunis 1002, Tunisia

**Keywords:** HPV infection, epidemiology, Tunisian women, vaccine

## Abstract

There are limited national population-based studies on HPV genotypes distribution in Tunisia, thus making difficult an assessment of the burden of vaccine-preventable cervical cancer. In this context, we conducted a national survey to determine the HPV prevalence and genotypes distribution and the risk factors for HPV infections in Tunisian women. This is a cross-sectional study performed between December 2012 and December 2014. A liquid-based Pap smear sample was obtained from all women and samples’ DNAs were extracted. Only women with betaglobin-positive PCR were further analysed for HPV detection and typing by a nested-PCR of the L1 region followed by next-generation sequencing. A multiple logistic regression model was used for the analysis of associations between the variables. A total of 1517 women were enrolled in this study, and 1229 out of the 1517 cervical samples were positive for the betaglobin control PCR and tested for HPV. Overall HPV infection prevalence was measured to be 7.8% (96/1229), with significant differences between the grand regions, ranging from 2% in the North to 13.1% in Grand Tunis. High-risk HPV genotypes accounted for 5% of the infections. The most prevalent genotypes were HPV 31 (1%), 16 (0.9%), 59 (0.7%). HPV18 was detected only in four cases of the study population. Potential risk factors were living in Grand Tunis region (OR: 7.94 [2.74–22.99]), married status (OR: 2.74 [1.23–6.13]), smoking habit (OR: 2.73 [1.35–5.51]), occupation (OR: 1.81 [1.09–3.01]) and women with multiple sexual partners (OR: 1.91 [1.07–3.39]). These findings underscore the need to evaluate the cost effectiveness of HPV vaccine implementation, contribute to the evidence on the burden of HPV infections, the critical role of sexual behaviour and socioeconomic status, and call for increased support to the preventive program of cervical cancer in Tunisia.

## 1. Introduction

Cervical cancer (CC) is the fourth most common cancer among women with an estimated 570,000 new cases in 2018 and the death cause of an estimated 311,000 women worldwide each year—90% of whom live in low- and middle-income countries [1]. In Tunisia, CC ranks second among gynaecological cancers of women after breast cancer, with an incidence of 5.8 per 100,000 women, or 250 to 300 new cases per year [1].

CC is linked to genital Human papillomavirus (HPV) infection, which is the most common sexually transmitted infection in the world [2]. Persistent infection with high-risk HPV (HR-HPV) can lead to precancerous lesions that are likely to progress into cervical cancer [3]. Most common oncogenic HPV genotypes are HR-HPV16 and 18 that have been identified as the most prevalent in CC samples in previous Tunisian reports [4,5].

Different types of vaccines against HPV infections are currently available worldwide: two bivalent vaccines targeting HR-HPV: HPV 16 and 18 (Cervarix and Innovax [6]); a Quadrivalent vaccine targeting HR-HPV 16 and 18 and low risk HPV (LR-HPV): HPV 6 and 11; and a Nona-valent HPV vaccine protecting against seven HR-HPV genotypes (HPV6, 18, 31, 33, 45, 52, and 58) and 2 LR-HPV genotypes (HPV6, 11) [7,8]. They have the potential, in complementarity with screening, to reduce the incidence of CC. In the light of the available data and advances in the development of HPV vaccines around the world, Tunisia is facing the need to discuss the benefits of introducing these vaccines. The implementation of a public health preventive strategy of CC including vaccination in Tunisia requires a preliminary epidemiological study of the prevalence of HPV infections and the circulating viral genotypes.

The previous published studies that have estimated the prevalence of HPV in Tunisia were limited to specific study populations or to specific regional areas and have used different tests and methods [4,5,9,10,11,12,13,14].

This is the first national study that aimed to determine the HPV prevalence and distribution, to identify the most common circulating genotypes, and to assess the risk factors for HPV infection in Tunisian women.

## 2. Materials and Methods

### 2.1. Type and Areas of Study

This is a national cross-sectional descriptive study performed between December 2012 and December 2014 and conducted by the National Observatory of New and Emerging Diseases in collaboration with the Institut Pasteur de Tunis (EMRO WHO HPV reference laboratory). Samples were collected from the 24 governorates of Tunisia in health centres (Appendix A). The governorates were regrouped into regions and regions regrouped into grand regions (Appendix A).

### 2.2. Study Population

The number of subjects to be included in the study (*n*) was calculated using the formula for a simple random sample (*n* = DE (zα/22p (1 − *p*))/d2)), depending on the desired accuracy (d) of 2%. HPV infection prevalence was estimated to *p* = 8% with α error risk of 5% (z_α_/2 = 1.96). For best accuracy we took a correction factor dE = 1.5 [10].

The inclusion criteria were women aged 18–75 years, attending [local primary health care centres [Centre de soin de base, herein referred to CSB] or the regional reproductive health centres [Centre régional de santé de reproduction, herein referred to CRSR] during the study period, who were sexually active, and who gave written consent. The two types of health centres are considered as first line healthcare in Tunisia. Selection of CSBs and CRSRs was made proportionally to the size of the governorates’ women population. Women having a medical history of cervical surgery or/and women with unsatisfactory cervical smear (according to Bethesda system) or samples with poor DNA quality, were excluded from the study.

### 2.3. Samples and Data Collection

Trained regional teams composed of medical doctors and midwives conducted face-to-face interviews using a standardized questionnaire, to collect information on socio-demographic data, socio-economic status, medical history, reproductive status, sexual behaviour of the woman and her partner. Once informed consent was signed, a liquid-based cervical sample was obtained using a cervix brush (Cervexbrush, Easyfix solution Labonord, Templemars, France). Samples were kept at 4 °C and sent to the Laboratory of Human and Experimental Pathology at the Institut Pasteur de Tunis within a week. Part of the sample was used for a Pap smear and the remaining content stored at −20 °C for further DNA extraction.

### 2.4. Cytological Diagnosis

Cytological diagnosis was blindly performed at the laboratory of Human and Experimental Pathology at the Institut Pasteur de Tunis and the conclusion was taken by consensus according to the Bethesda system.

### 2.5. DNA Extraction

DNA was extracted from a 200 μL aliquot of the suspended cell samples using the QIAamp DNA Blood Mini Kit (Qiagen, Hilden, Germany) according to the manufacturer’s instructions. The quality of extracted DNAs was evaluated by the β-globin-specific PCR primers, PC04/GH20. Only β-globine PCR positive samples were further used for HPV detection and genotyping.

### 2.6. HPV Detection and Genotyping

HPV detection was performed by a nested PCR using biotinylated PGMY09/11 primers for the first PCR and GP5+/GP6+ for the second PCR. For HPV genotyping a protocol based on Next Generation sequencing (NGS) was performed as reported previously [15]. Briefly, for libraries production, a nested PCR was performed using in the first PCR PYGMY primers followed by a second PCR with modified GP5+/GP6+ primers (IP5-GP5+/IP7-GP6+) [15]. The obtained amplicons were used for adding indexes. Libraries were purified, quantified then submitted to Miseq sequencing [15]. For data analysis generated reads were mapped to reference human genome using Bowtie tool. Only unmapped reads were processed to assembly using Trinity software (https://github.com/trinityrnaseq/trinity, accessed on 8 August 2019). To identify HPV genotypes, obtained contigs were clustered and aligned using Bowtie 2 with the L1 region of alpha Papillomavirus reference sequences that were obtained from Papillomavirus Episteme (PaVE) (PaVE; https://pave.niaid.nih.gov/, accessed on 15 September 2022) [15].

### 2.7. Statistical Analysis

Qualitative variables were described by the simple counts and percentages, quantitative variables by the mean ± Stand Error of Mean (SEM). The distribution of HPV genotypes was summarized using frequency distributions. The prevalence of HPV infection, presence of single or multiple HPV infections, and HPV genotypes, as well as their corresponding 95% CIs were estimated with binomial distribution analysis. The Chi-square test was performed to compare the differences in the prevalence of HPV or HR-HPV between sub-groups of subjects.

Demographic, epidemiological, and clinical risk factors of HPV positivity were first examined by univariate analysis using the two-sample test for normally distributed continuous data, the Mann–Whitney test for non-parametric data, the chi-squared test for frequencies and the Fisher’s test and the chi-square test for trends. To identify the associated risk factors of HPV infection, only statistically significant variables (*p* < 0.05) and those with *p* < 0.2, from univariate analysis, were left in multivariate analyses. Multivariate analysis was performed using backward stepwise model. Odds Ratios (ORs) and 95% confidence intervals (CI) were used to quantify the association between risk factors and positivity for HPV. A “two side” *p*-value < 0.05 was considered statistically significant. Data analysis was performed using SPSS version 22 software.

## 3. Results

### 3.1. Demographic and Cytological Description of the Study Population

The study involved 1517 women (Appendix A). Two hundred eighty-eight women were excluded because of insufficient cellularity, a suspicion of contamination affecting the samples during one of the various steps of manipulation (cytology, DNA extraction), coding error or negative β-globin PCR (Figure 1). In total, 1229 β-globin PCR positive samples were defined as of good quality and were therefore tested for HPV detection and testing, and so were eligible for the statistical analyses (Figure 1).

Of note, no significant statistical differences were observed between excluded (negative β-globin PCR) and included women in this study (Appendix A).

The mean age of participants was 40 years ± 9.2 and 54% were between 30 and 40 years old. Most of the participants were married (94.5%), 20.4% had no formal education, 70.7% were unemployed and only 5.9% had never smoked. About 34.3% of women in this study had sexual activities before 20 years old and 15.2% had multiple sexual partners (Table 1).

Cytological study revealed that 612 women (49.8%, 95%CI [47.02–52.59]) had normal cytology and only 85 women (6.9%, 95%CI [5.62–8.47]) presented a cytology with intraepithelial lesions (Figure 2).

The distribution of cytological profiles of the study population is analyzed. Histogram reporting the distribution of different cytological profiles identified in this study is presented. Cytology with benign transformations regrouping inflammatory and atrophic cytology is studied, as is cytology with intraepithelial lesions regrouping low grade and high-grade intraepithelial lesions.

### 3.2. Global HPV Prevalence and Genotypes Distribution

Among the 1229 β-globin PCR-positive samples, HPV was detected in 96 women. Thus, the national prevalence of HPV infection in Tunisia is estimated at 7.8% (96/1229; 95% CI [6.5−9.4]). 

Multiple infections were detected in only 21.9% of the HPV positive samples (21/96) and single infections in 78.1% (75/96). The number of genotypes found in multiple infections ranged from 2 to 5 (Table 2).

A total of 25 HPV genotypes were identified in this study. The prevalence of HR-HPVs and LR-HPVs genotypes were about 5% (60/1229; 95% CI [3.88–6.33%]) and 4.1% (50/1229; 95% CI [3.17–5.41%]), respectively. The most common HR-HPVs were HPV31 (1% (12/1229); 95% CI [0.56–1.70%]), HPV16 (0.9% (11/1229); 95%CI [0.50–1.59]), HPV59 (0.7% (8/1229); 95% CI [0.33–1.27%]) and HPV52 (0.7% (7/1223; 95% CI [0.37–1.57%]. HPV6 (1.6% (20/1229); 95% CI [1.06–2.50%) and HPV40 (1.1% (13/1229); 95% CI [0.62–1.80%]) were the most prevalent LR-HPV. The prevalence of the detected HPV genotypes is illustrated on Figure 3.

### 3.3. HPV Infection and Genotypes Distribution According to the Cytology Status of the Studied Samples

HPV infections were more prevalent in smears with intraepithelial lesions (11.8%), but no significant association was observed between cytological status and HPV infection (Table 3).

HR-HPV genotypes were mostly observed in smears presenting normal cytology (28/49) or having benign transformation (15/37) (Figure 4).

HPV16 (14%) and HPV18 (4%) were mainly detected in smears with normal cytology (Appendix A). In a decreasing order, HPV31 (16.7%), HPV59 (12.5%), HPV16 (11.1%) and HPV68 (8.3%) were the most HR-HPV genotypes detected in cytology with benign transformations (Appendix A). Only 4 HR-HPV genotypes (HPV31, HPV52, HPV53 and HPV68) were identified in cytology with intraepithelial lesions. Prevalence of each HPV genotype according to cytological profiles is detailed in Appendix A.

### 3.4. HPV Infection and Distribution According to Age

The mean age of HPV-positive women was 38.39 ± 1.019 years. HPV prevalence did not vary significantly among the different age groups. However, two prevalence peaks were found: the first in less than 30 years old group with a prevalence of 12.4% (25/202, 95%CI [8.4–36.9%]); the second for women over 50 (8.1% (13/160), 95%CI [6.8–19.6%]) (Figure 5).

Multiple infections were mainly found in young women (under 30) (24%) and women aged between 40 and 50 years old (37%) over 40 years of age. However, this observation was not statistically significant (*p* = 0.08) (Table 4).

HR-HPV infections increased significantly with age and were more prevalent among women with age bellow 30 years old or over 50 years old (Figure 5). We measured a significant difference in the variation in prevalence by age group in HR-HPV infections (*p* = 0.002) but the difference was not significant for LR-HPV (*p* = 0.45).

### 3.5. HPV Infection and Distribution According to Tunisian Regions

The prevalence of HPV infection by region is shown in the figure below. The highest prevalence was observed in the regions of Grand Tunis (13.1%; 95% CI [2.8–14.6%]) and South-East (13%; 95% CI [7.5–21.2%]. However, the lowest was observed in the North-East (1.4%; 95% CI [0.6–5.1%]) (Figure 6). This difference was statistically significant (*p* < 10^−3^).

When we regrouped the different regions to grand-regions; South-East and South-West to South, North-East and North-West to North and Central-East and Central-West to Central, the results remained the same. The lowest value was observed in the grand-region of the North (2%; 95% CI [0.91–4.25] and the highest in the grand-region of Grand Tunis (13.1%) (95% CI [2.8–14.6%]). The regions of South and Central presented a prevalence of 11.6% (95% CI [7.6–17.2]) and 6.3% (95% CI [4.4–9]), respectively.

### 3.6. Risk Factors Associated to HPV Infection in the Tunisian Population

For the univariate and multivariate analysis, we considered the grand regions: South-East and South-West to South, North-East and North-West to North and Central-East and Central-West to Central.

In a univariate analysis, the socio-demographic factors associated with HPV infection were marital status, professional activity, and geographical area (grand-region). Medical history factors significantly associated to HPV infection were history of sexually transmissible infections and parity. Significantly associated behavioral factors were smoking and multiple partners of the women.

Age, educational level, and occupation have not shown a significant link with HPV infection. Nevertheless, from the univariate analysis, their *p*-values did not exceed 0.2 (Appendix A) which did not exclude them from the multivariate analysis.

Multivariate analysis suggests that living in Grand Tunis region, having married status, professional activity and multiple partners, and smoking were independently associated with HPV infection. However, the age at sexual intercourse debut and the increasing number of sexual encounters of the partner were not significant in the multivariate model (Table 5).

## 4. Discussion

The majority of CC cases occur in poor countries, with Latin America, Asia, and Africa having the greatest rates [16]. In Tunisia, CC is the second gynaecological malignancy after breast cancer [1]. Regarding the poor efficiency of pap screening preventive strategy and about 400 new cases each year, CC represents a major health issue in Tunisia [17]. Up to now, the data on women HPV cervix infection prevalence and HPV genotype distribution in Tunisia was limited to some governorates and related to specific categories of women [4,9,10,11,12,13]. Therefore, there was a need for a national investigation on HPV epidemiology to obtain a detailed picture of the prevalence and distribution of HPV genotypes in Tunisian women population. In this context, this study aimed to evaluate HPV prevalence, genotype distribution and associated risk factors in Tunisian women at a national level.

The national prevalence of HPV cervix infection was 7.8%. The highest prevalence was seen in the grand regions of Grand Tunis (13.2%) and South (13%) while the lowest was observed in the North (2%). This difference was statistically significant. The disparity in HPV distribution could be linked to sexual behaviours and differences in levels of conservative behaviour in the Grand Tunis and North. This latter region could be considered as mostly rural, and it is known that rural communities are typically more conservative than urban communities. Different studies from diverse geographical areas/countries reported that HPV infection prevalence was higher in urban than rural areas [18,19,20,21].

According to previous reviews and meta-analysis, the adjusted global prevalence of HPV infection was estimated at 11.7% and the country-adjusted HPV prevalence ranged from 1.6% to 41.9% worldwide. It was reported that the highest prevalence of HPV infection was observed in some parts of sub-Saharan Africa (24%). Latin America and the Caribbean (16.1%) [22,23], in Eastern Europe (14.2%) and in South-East Asia (14%) [23]. In the European continent, the highest HPV prevalence of infection was found in Italy and Spain, where about 5% of the general population is infected [24]. Data concerning North African countries and Egypt reported a rate of 10.5%, which is very close to our data in some governorates [25,26,27]. These variations seem to be related to the studied populations, the sample size, the geographical area, the sampling methodology, the data collection methods, and the tests used.

Regarding the age of infected women, we found two peaks: women under 30 (20.0% [8.4–36.9%]) and over 50 years (17.0% [6.8–19.6%]. The second peak observed in our study has been described in most developed countries [28,29,30,31,32]. The appearance of the first peak depends on the beginning age of sexual relations. Overall, the trend where HR-HPV infection shows high rates in younger groups and low rates in middle age groups reflects the natural evolution of immune system responses. Young women are more likely to clear the viruses within one or two years in 70% to 91% of cases [33,34]. The slight increase of HPV infection rate in older women might reflect the viral persistence or reactivation of latent HPV infection(s), likely because of the physiological and immunological disorders that can result from hormone fluctuations during the menopausal transition [35].

The HPV genotypes distribution obtained in this study agrees with the widely reported data, clearly showing that HPV16, 31, 59 and 52, the most oncogenic HPVs, are the prevalent genotypes in Tunisia. This finding may encourage the introduction of the Nona-valent HPV prophylactic vaccine that preferentially targets these HR-HPVs genotypes [7]. One important thing must be however noticed and taken into consideration is that so far HPV16 and HPV18 were the most detected viruses in CC in previous Tunisian studies [5]. This means that HPV18 could be less prevalent but is the second most oncogenic HR-HPV genotype.

From a cytological point of view, HPV was identified in only 11.8% of samples with precancerous lesions. This can be due to the low specificity of Pap smears reading or to a low viral charge, knowing that the viral charge decreases with the lesion progression [36,37]. On the other hand, HPV was found in 15% of cases having normal smears (8%) or with benign modifications (7%). This is a proof of the lack of sensitivity of pap smears for diagnosis and screening. Considering the HR-HPV genotypes detected in precancerous lesions, only 4 HR-HPV genotypes were detected (HPV31, HPV52, HPV53 and HPV68). Surprisingly, HPV16 and HPV18 were mainly observed in normal smears with a rate of 14% and 4%, respectively. This is not concordant with previous reports where HPV 16 was the most commonly detected type in precancerous and cancerous cervix lesions [38]. However, it adds to the findings of a recent Chinese study that showed that HPV16 and HPV58 were the most prevalent HPV genotypes in cervical lesions [39]. This difference could be explained by the fact that genotypes other than HPV16 were the most frequently found in Tunisian women population. In our sample, 10.5% of women had smears with low grade lesions and 1.0% had smears with high grade lesions and smears having atypical glandular cells of undetermined significance (AGUS). Although no epidemiological study of this magnitude has so far investigated the distribution of genotypes in this type of lesions in Tunisia, we cannot compare our results to other national and international data since we are limited by the low number of women with cervical lesions in this study.

As regards to risk factors, the multivariate model used in our study population retained geographical region, occupation, marital status, smoking and sexual habits of women as potential risk factors by adjusting for other potential confounders. According to the literature, the main risk factors associated with HPV infection in women are age, socio-demographic characteristics, sexual behaviour, history of sexually transmitted infections, parity, contraceptive methods, and smoking [40,41,42]. Many studies have provided several factors that negatively or positively affect virus transmission or persistence. Age at the time of first sexual contact is a factor that is less often identified. Other factors such as history of STIs, hormonal factors (oral contraceptives and pregnancies), use of condom are sometimes associated with HPV infection [21,40,42,43]. The differences found in the present study are likely due to our population specificities. Although it is possible that women found it difficult to answer questions related to their sexual life, the total number of sexual partners and the number of recent partners appeared to be the most consistent determinants, particularly for oncogenic HPV infection.

## 5. Conclusions

This study reports a 7.8% HPV prevalence in Tunisian women with significant differences between regions. A high rate of HR-HPV infection has been identified. HPV16 and 31 were the most prevalent, whereas HPV 18 was detected in only four cases. It is important to take into consideration that HPV18 was one of the most frequent in invasive cervical carcinoma and despite its little prevalence in this study remains one of the most oncogenic genotypes. These findings are important for the update and the optimization of preventative strategies against HPV infections and cervical cancer in Tunisia. The introduction of the Nona-valent vaccine could strengthen the prevention of cervical cancer in Tunisia. Its implementation should be discussed considering its cost-effectiveness and the targeted population in relation to circulating HPV genotypes in Tunisia.

## Figures and Tables

**Figure 1 viruses-14-02175-f001:**
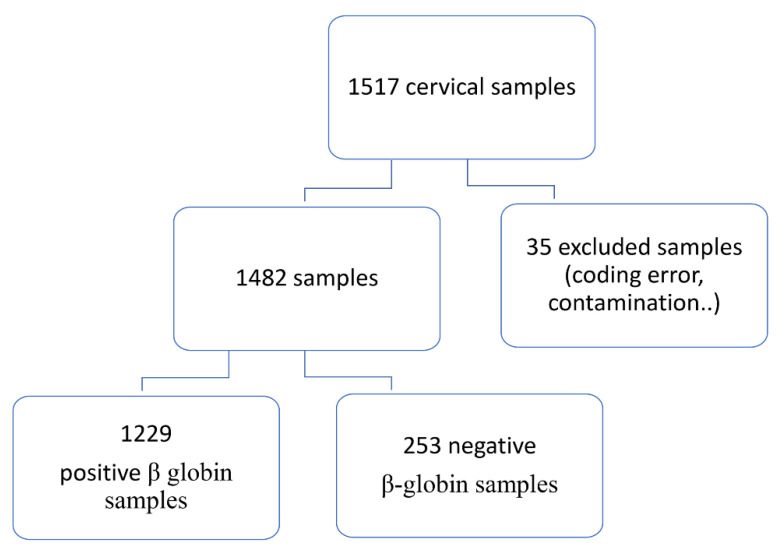
Flow chart of the study samples.

**Figure 2 viruses-14-02175-f002:**
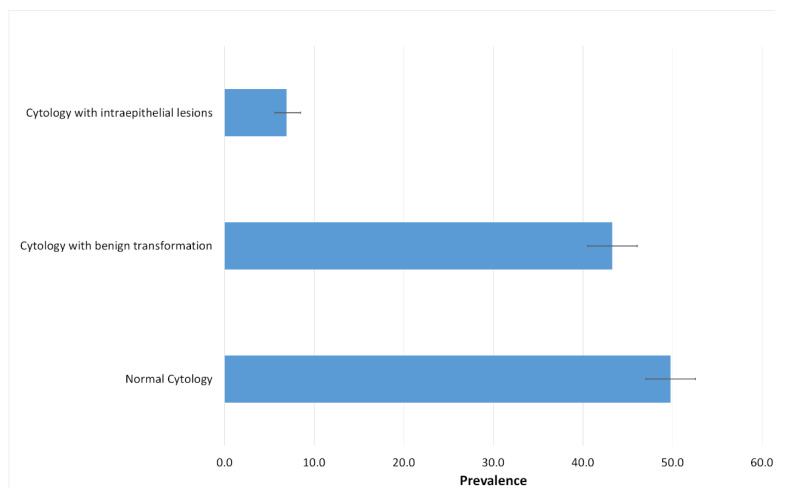
Distribution of cytological profiles of the study population.

**Figure 3 viruses-14-02175-f003:**
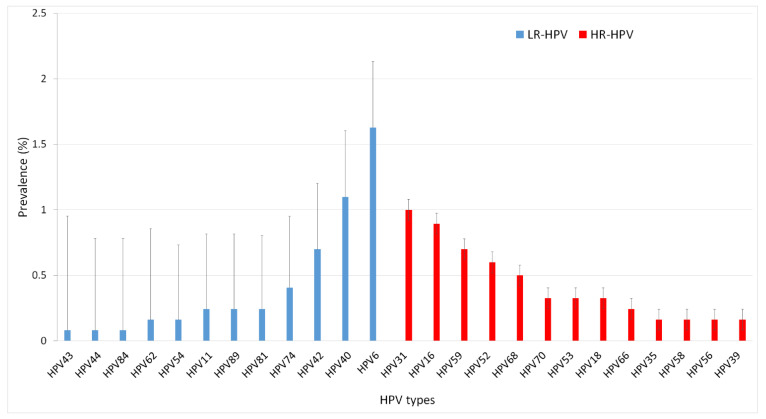
High-risk and Low-risk HPV genotypes prevalence in Tunisian population. HPV16 (1%) and HPV31 (0.9) are the most prevalent high-risk HPV genotypes. HPV70, HPV53 and HPV18 occupied the 8th position, with a prevalence of 0.33%. HPV6 1.6%) and HPV40 (1.1%) are the most prevalent low risk HPV genotypes followed by HPV42 (0.57%), HPV74 (0.41). HPV81, HPV89 and HPV11 are presented with a prevalence of 0.24%.

**Figure 4 viruses-14-02175-f004:**
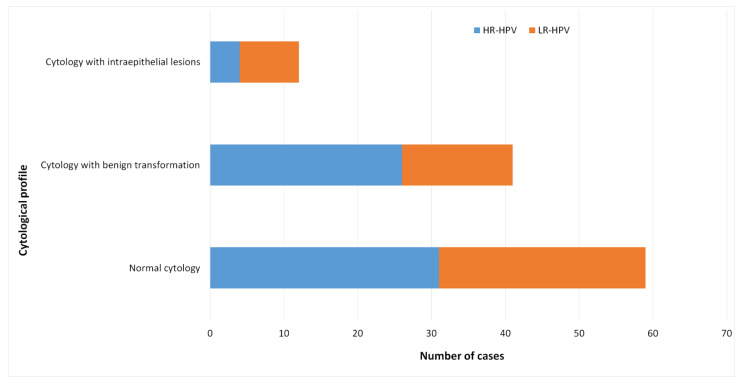
High risk and Low risk HPV genotypes distribution according to the cytological profiles. Histogram reporting the number of samples with high-risk HPV (HR) and low risk HPV (LR) according to the cytological profile.

**Figure 5 viruses-14-02175-f005:**
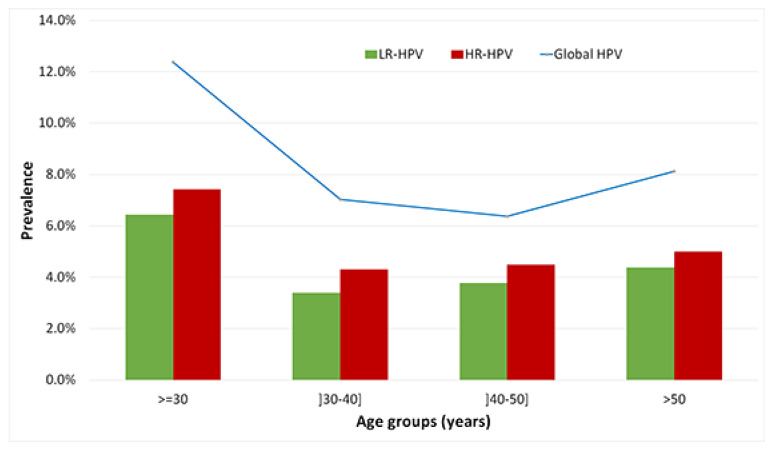
Global HPV Prevalence and High risk and Low risk HPV genotypes distribution according to age-groups. Blue line represents the distribution of global HPV prevalence according to age groups. Histograms represent the distribution of high-risk HPV (HR) and low risk HPV (LR) genotypes according to age groups.

**Figure 6 viruses-14-02175-f006:**
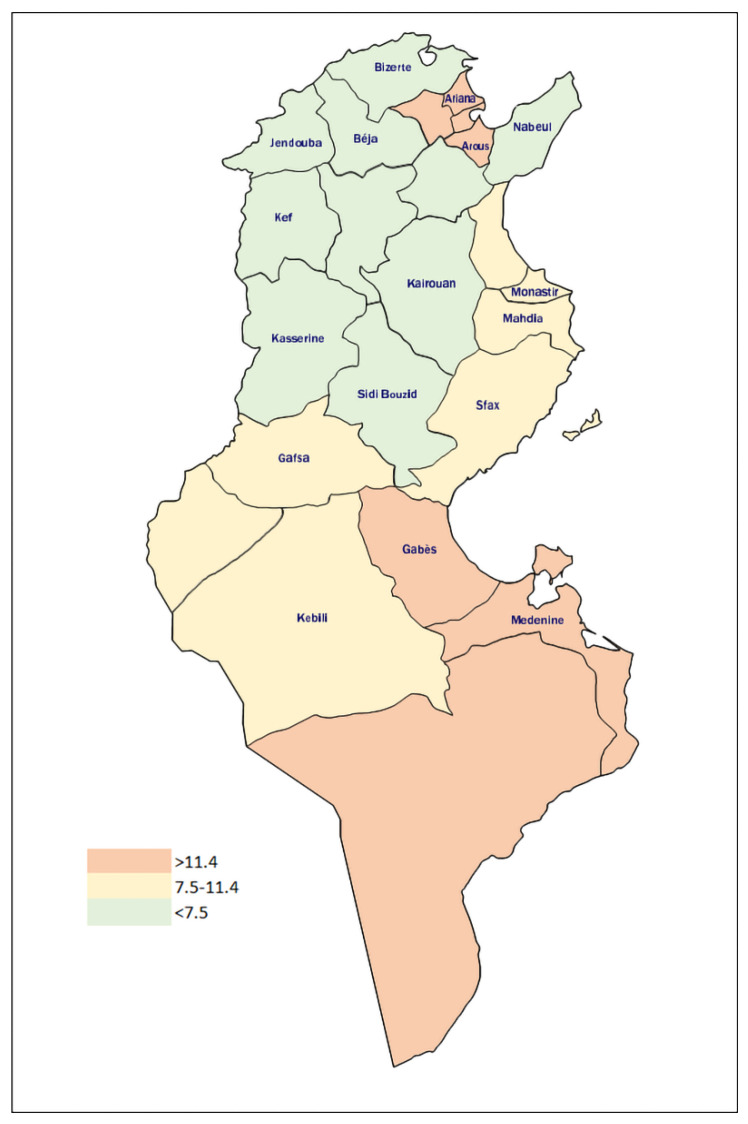
HPV distribution according to the Tunisian regions. The highest prevalence (orange colour) is observed in the regions of South-West (Kebili, Gafsa and Tozeur governorates) and Grand Tunis (Manouba, Tunis, Ariana and Ben Arous) with a prevalence of 13% and 13.1% respectively. The North-East (Nabeul, Zaghouan and Bizerte), North-West (Beja, Jendouba, Kef and Siliana) and Central-West (Kairouan, Kasserine and Sidi Bouzid) regions showed the lowest (Green colour) prevalence of HPV (2.4%, 2.2% and 2% respectively). Central East (Sousse, Mahdia, Monastir and Sfax) and South-West (Gafsa, Tozeur and Kebeli) regions presented a prevalence of 8.6% and 8.3%, respectively.

**Table 1 viruses-14-02175-t001:** Socio-demographic characteristics of the study population (N = 1229).

Variable	*n*	%
Regions		
Grand-Tunis	327	26.6
North-East	164	13.3
North-West	139	11.3
Central-East	280	22.8
Central-West	147	12
South-East	100	8.1
South-West	72	5.9
Collection centre		
CSB	642	52.2
CRSR	587	47.8
Age groups *		
<30	202	16.5
30–40	441	36.0
40–50	423	34.5
>50	160	13.1
Marital status		
Single	68	5.5
Married	1161	94.5
Level of education *		
Illiterate	307	20.4
Primary	599	39.7
Secondary and high level	602	39.9
Occupation		
Yes	360	29.3
Non	869	70.7
Housing type		
Traditional house	530	43.1
Villa	570	46.4
Apartment	110	9.0
Rudimentary	19	1.5
Monthly household income (TND) *		
<350	420	34.2
350–700	463	37.7
>700	298	24.2
Smoking		
Yes	72	5.9
No	1157	94.1
Medical history *		
Yes	367	30.1
No	854	69.9
Surgical history *		
Yes	446	36.4
Non	778	63.6
Menopause *		
Yes	235	19.3
No	985	80.7
Pregnancy *		
Yes	82	7.4
No	1029	92.6
Contraception *		
Yes	699	59.5
No	475	40.5
Gestality *		
0	45	3.7
>1	1171	96.3
Parity *		
0	69	5.7
1, 2	436	36.2
>3	701	58.1
Age at first intercourse		
<20	394	34.3
≥20	755	65.7
History of STI *		
Yes	240	19.7
No	977	80.3
Number of partners *		
unique partner	941	84.8
multiple partners	169	15.2
Multiple sexual intercourse of partner *		
Yes	79	6.6
No	1100	91.7
Not sure	21	1.8

*: N is different from 1229 due to missing data, STI: sexual transmission infection.

**Table 2 viruses-14-02175-t002:** Distribution of single and multiple HPV infections.

	Single Infection	Multiple Infection				
		Double	Triple	Quadruple	Quintet	Total
*n* (%)	75 (78.1%)	12 (12.5%)	7 (7.3%)	1 (1%)	1 (1%)	96 (100%)

**Table 3 viruses-14-02175-t003:** HPV infection status according to cytology.

	HPV Positive	HPV Negative	*p*
Normal cytology	49 (8%)	563 (92%)	0.25
Cytology with benign transformations	37 (7%)	495 (93%)	
Cytology with intraepithelial lesions	10 (11.8%)	75 (88.2%)	

*p*: *p* value for the association between HPV status and cytological profile.

**Table 4 viruses-14-02175-t004:** Single and multiple infections distribution according to age groups.

Age-Groups (Years)	Simple Infection N (%)	Multiple Infections N (%)	*p*
<30	19 (76%)	6 (24%)	0.08
(30–40)	28 (90.3%)	3 (9.7%)	
(40–50)	17 (63%)	10 (37%)	
>50	11 (84.6%)	2 (15.4%)	

*p*: *p* value for the association between HPV type infection and age groups.

**Table 5 viruses-14-02175-t005:** Multivariate analysis of associated factors with HPV infection in Tunisian women.

Variable	AOR	*p*
Regions		
Grand Tunis	7.94 [2.74–22.99]	<0.0001
South	5.37 [1.55–18.58]	
Central	4.01 [1.33–12.08]	
North	Ref	
Marital status		
Married	2.74 [1.23–6.13]	0.014
Single (widow, divorced, never married)	Ref	
Occupation		
Yes	1.81 [1.09–3.01]	0.022
No	Ref	
Smoking		
Yes	2.73 [1.35–5.51]	0.005
No	Ref	
Number of sexual partners		
Multiple partner	1.91 [1.07–3.39]	0.027
One partner	Ref	

Ref: Reference category, AOR: Adjusted Odds Ratio.

## Data Availability

Data is contained within the article or Appendix A.

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
