# Peer review of "The Prevalence, Genotype Distribution and Risk Factors of Human Papillomavirus in Tunisia: A National-Based Study"

_viruses, 2022, doi:10.3390/v14102175_

Round 1
Reviewer 1 Report
1. Tables and figures are unclear
2. Below the pictures there should be an explanation of the picture so that it is completely understandable to the reader by itself (self-contained)
For example picture 5 is completely incomprehensible and it cannot be understood even by reading the text (line 201-204)
3. In the pictures, put values on the columns, so it might be easier to understand what the authors wanted to show
4. For the abbreviations CSB and CRSR, what the abbreviation means should be stated in brackets
5. English should be reviewed again, e.g
Line 141 - ...and were therefore tested for HPV detection and testing
Line 197-8 ...women aged between 40 and 50 years old (37%) over 40 years of age.
Line 269, 317 ... HPV cervix infection - HPV cervical infection.....cervix lesions - cervical lesions
6. There are many errors in the numbers, e.g
Line 172 .. HPV59 (0.7% (8/1229) and HPV52 (0.7% (7/1223)
Line 193 ...the second for women over 50 (8.1% (13/106), 13/160
Line 288-9.....Regarding the age of infected women, we found two peaks: women under 30 (20.0% [8.4% -36.9%]) and over 50 years (17.0% [6.8% -36.9%]). 19.6%]. These data in the discussion are not consistent with those in the results.
7. This subtitle doesn't look good: 3.1. Prevalence measures of the infection at the national level and of the HPV genotype
8. Line 311 ....lack of specificity of pap smears for diagnosis and screening..... lack of sensitivity of pap smears....
Reviewer 2 Report
This study aimed to evaluate HPV prevalence, genotype distribution and associated risk factors in Tunisian women at a national level. The results are important for Tunisia given the lack of national data and the lack of national HPV-vaccination program.
This study is cross-sectional and well performed given the restrictions of cross-sectional data. The data is quite old (data from 2012-2014). The manuscript is well-written, easy to read and easy to understand.
I have a few minor comments
1. There seems to be typos in line 77 (genotypes should be types?) and in line 346 (warlike).
2. I don´t understand the statement "Multiple sexual intercourse of partner" in the questionnaire of which 92% stated "no".
